# A Situation-Specific Theory of End-of-Life Communication in Nursing Homes

**DOI:** 10.3390/ijerph20010869

**Published:** 2023-01-03

**Authors:** Silvia Gonella, Sara Campagna, Valerio Dimonte

**Affiliations:** 1Direction of Health Professions, City of Health and Science University Hospital of Torino, Corso Bramante 88-90, 10126 Turin, Italy; 2Department of Public Health and Pediatrics, University of Torino, Via Santena 5 bis, 10126 Turin, Italy

**Keywords:** case study, communication, development, end of life, nursing home, situation-specific theory, theorizing process, theory

## Abstract

High-quality end-of-life communication between healthcare professionals (HCPs), patients and/or their family caregivers (FCs) improves quality of life and reduces non-beneficial care at the end of life. Nursing homes (NHs) are among the contexts at the forefront of these conversations. Having a solid theoretical basis for the role of end-of-life communication in NHs in transitioning to palliative-oriented care can offer indications for research, practice, education, and policy related to geropalliative care. This study aimed to develop a situation-specific theory of end-of-life communication in NHs by refining an existing theory. A four-step integrative approach was employed that included: (1) checking the assumptions for theorization; (2) exploring the phenomenon through multiple sources; (3) theorizing; and (4) reporting. All elements of the existing end-of-life communication theory in NHs were confirmed: end-of-life communication improved the understanding of FCs about their relatives’ health conditions, shared decision-making, and reflections on the desired preferences of residents/FCs for care at the end of life. Furthermore, the family environment affected the burden of FCs in the decision-making process. Finally, time and resource constraints, regulations, visitation restrictions due to the COVID-19 pandemic, and social and cultural values influenced the quality and timing of communication. The study findings confirmed the impact of the political, historical, social, and cultural context on end-of-life communication, thus providing the basis for a situation-specific theory.

## 1. Introduction

Among people over 65 years old, one in five deaths occurs in nursing homes (NHs) with the percentages doubling with each 10-year increase in age [1]. Almost 90% of people aged 85 years or older die in NHs and NH deaths are projected to double by 2040 [2]. People usually transition into a NH when their clinical conditions and cognitive capacity have already been severely compromised, and their life expectancy is poor [3]. In Italy, three-quarters of the residents who die in NHs have more than two morbidities, almost 80% suffer from severe to advanced dementia, and the median length of stay from admission to death is 14 months [4]. Thus, family caregivers (FCs) often assume a surrogate decision-making role [5] despite often not knowing their relative’s preferences with certainty, and their stress increases as their relative approaches the end of life [6]. Therefore, healthcare professionals (HCPs) should offer FCs ongoing and sensitive end-of-life communication to provide emotional support and improve the quality of FCs’ remaining time with their relative [7]. FCs indeed experience a high quality NH environment when HCPs provide individualized attention, are responsive to their needs, and are open to dialogue [8,9].

Communication has been set among the international palliative care research priorities, since high-quality, interactive communication about prognosis and care preferences may enable patients and FCs to prepare for approaching death, and has the potential to improve the acceptance of a prognosis based on in-depth understanding and to align treatments with their desires [10]. When end-of-life communication takes place, non-beneficial care decreases and the quality of dying and satisfaction with care improve [11]. Instead, a lack of end-of-life communication results in a progression toward death with a high symptom burden and a decreased quality of life for both patients and families [12].

In Italian NHs, more than half of residents receive at least one potentially inappropriate treatment in the last week of life [13]. Specifically, approximately 5% of residents receive at least one critical care treatment (e.g., resuscitation, artificial ventilation) and about 40% receive at least one artificial nutrition or hydration treatment [13]. One in six Italian residents visit the emergency department in the last month of life and over 80% of them die in the hospital [4]. End-of-life conversations between the HCPs and a relative of the resident about the preferred medical treatments and the course of care in the last phase of life were found to limit life-sustaining treatments and hospitalization [11,14]. When the discussion occurred, the FCs generally preferred to avoid burdensome hospitalizations and leave their relative to die quietly in the NH [9]. Unfortunately, end-of-life communication in NHs is often delayed and poor, and FCs are not informed about treatment options and are not prepared for their role as a decision-maker if their relative loses cognitive capacity [15]. Less than 40% of the FCs of NH residents with advanced dementia remembered any communication about the possible treatment options for their relative at the end of life [16], only one in five joined care plan meetings [17], and no discussion occurred in over one-third of cases [18]. Similarly, less than 60% of Italian FCs remembered conversations with the NH staff about the preferred care in the last month of their relative’s life [14].

Although national consensus bodies recommend making end-of-life communication part of the routine care for seriously ill patients and their families [19,20], the theoretical basis for the role of communication in transitioning towards palliative-oriented care at the end of life is still limited [21]. Solid theoretical bases on how end-of-life communication may work could connect theory with research and practice, and can help guide and sustain the development of interventions aimed at improving the communication skills of HCPs, particularly at the end of life, and offer indications for policy, education, and research related to geropalliative care. Indeed, strong theoretical underpinnings are essential to develop knowledge and improve clinical practice [22,23].

When theories are described in terms of their level of abstraction, they are usually categorized into grand theories, middle-range theories, and situation-specific theories with decreasing levels of abstraction [24]. Situation-specific theories have been argued to be the best theories when exploring complex phenomena such as end-of-life communication, since they are limited to specific populations or to particular fields of practice, they can incorporate the diversities and complexities of phenomena that reflect clinical practice, and they consider the political, historical, social, and cultural contextual factors [24].

### Background

A theory of end-of-life communication and its role in contributing to the transition towards palliative-oriented care in NHs is already available [21]. According to this theory, the NH environment may influence the timing and quality of end-of-life communication that in turn impacts end-of-life care by affecting (a) the FCs’ understanding; (b) the shared decision-making between HCPs and residents/FCs; and (c) the knowledge of residents’ preferences and (d) FCs’ preferences for end-of-life care. Timely and thorough communication contributes to the transition towards palliative-oriented care by promoting family understanding, fostering shared decision-making between HCPs and residents/FCs, and improving the knowledge of residents’ and FCs’ preferences. Instead, when communication is delayed or poor, the provision of curative-oriented care is more likely due to FCs’ lack of understanding, no shared decision-making, and residents’ and FCs’ preferences being unknown (Figure 1) [21]. This theory considers only how the environment internal to the NH may affect end-of-life communication, while the role of contextual factors external to the NH is not taken into account. However, the political, historical, and socio-cultural context in which human beings live was found to affect individual processes that in turn elicit different communication patterns [25]. This suggests that communication is a highly context-sensitive phenomenon. Therefore, complementing the existing end-of-life communication theory with such contextual factors would provide more meaningful and comprehensive theoretical foundations to explore end-of-life communication and improve clinical practice. Based on the assumption that situation-specific theories are the best theories to explore complex phenomena such as end-of-life communication and that political, historical, social, and cultural factors are mandatory in the development of a situation-specific theory [26], an instrumental case study was performed to confirm that contextual factors influence the timing and quality of communication between NH staff and residents’ families. Therefore, this study develops a situation-specific theory on the role of end-of-life communication in contributing to the transition towards palliative-oriented care in NHs by refining an existing theory [21].

## 2. Methods

### 2.1. Instrumental Case Study

#### 2.1.1. Study Design

This instrumental case study was conducted in conjunction with a transnational quality improvement project that implemented structured family care conferences (FCCs) for the FCs of people with advanced dementia in a NH (primary study) between March and June 2021 [27].

The instrumental case study design refers to the inquiry of a particular case in a real-life, present-day setting (i.e., a NH) to provide insights regarding a particular issue (i.e., end-of-life communication) [28]. This study design is often used to test and refine an established theory [29].

#### 2.1.2. Primary Study

A transnational multidisciplinary implementation study, known as mySupport study, that involved a consortium of six countries (Canada, United Kingdom, Ireland, Italy, the Netherlands, and the Czech Republic), explored the benefits of structured FCCs associated with written information to support the FCs of NH residents with advanced dementia who have to make decisions about their relatives’ end-of-life care [27].

The FCC was a one-hour meeting that involved a trained nurse and the FC(s) significant to the resident. The meeting was also open to other HCPs involved in the resident’s care. The FCC was made up of four phases (preparing, conducting, documentation, and follow-up) based on the clinical practice guidelines for conducting family meetings [30]. In the meeting, a trained nurse discussed comfort care practices at the end of life with the aid of written resources that the FC(s) had received in advance and read. The FCCs were scheduled when the FCs came to visit their relative.

#### 2.1.3. Current Study

This instrumental case study was conducted in an urban, non-profit, Piedmontese NH (northwest Italy) that provides care to 106 residents who are located in five wards according to their care needs. There is neither a dedicated dementia care unit nor a specific protocol regarding advance care planning and family meetings.

The Italian long-term care sector has historically been characterized by complexity and fragmentation; it is considered a matter of regional competence and the central government provides only general guidelines [31]. The amount of time for care activities depends on regional regulations. In Piedmont, NH staffing is regulated by the DGR 45-4248/30 July 2012, which defines the amount of time for care activities provided by HCPs according to the residents’ care intensity (i.e., from 8 min/day for residents at low-care intensity up to 30–46 min/day for residents at high-care intensity). On average, in Piedmont, there are 6.4 FTE nurses for every 100 NH beds who must provide physical, psychosocial, and educative care [32].

In this study, end-of-life communication has been operationalized as structured FCCs, and the contextual factors external to the NH have been defined as laws, regulations, guidelines, historical events, and cultural and social values [33].

#### 2.1.4. Data Sources

The study draws on four data sources: (1) the residents’ clinical records; (2) the FCs’ self-administered, validated questionnaires; (3) the FCs, NH staff, NH manager, and research staff semi-structured interviews; and (4) in-the-field notes.

The NH manager identified the FCs of residents with advanced dementia and the members of the NH staff who were the most engaged in end-of-life communication. In total, 13 FCs were approached and 11 FCs (five daughters, four sons, and two nieces, all highly educated, with a mean age of 59 years (range 49–75)) joined the study. One FC declined and another dropped out shortly after recruitment due to their relative’s death. Three NH staff members were identified and all adhered to the study. The interview guide differed according to the time point (before the FCC or after the FCC) and the interviewee. Table 1 shows the main interview questions for the FCs, the NH staff, the NH manager, and the research staff. The interview guides were based on the experience of the experts in qualitative methodology and end-of-life care who were responsible for the transnational study [27]. The interviews had a mean duration of 20 min (range 7–45 min). In all, 79 pages of transcript were produced.

Data from the residents’ clinical records (*n* = 11) whose FC(s) adhered the study were also collected. Data collected from the residents’ clinical records referred to the 12 weeks before and the 12 weeks after the FCC and included (i) community-based services used, (ii) hospital services used, and (iii) completed documents (e.g., advanced care planning, decision to refuse treatments).

The FCs filled in the Decisional Conflict Scale [34] and the Family Perception of Care Scale [35] just before the FCC and 6 weeks after (*n* = 11 each). The former is a 16-item scale that measures uncertainty and difficulties in the decision making process (5-point Likert scale, 0 = no conflict; 4 = maximum conflict); the latter is a 25-item scale that explores the FCs’ perception of the care provided to their relative, including appropriate placement in the NH, the support they received during the decision-making process, and the quality of communication (7-point Likert scale, 1 = extremely bad; 7 = excellent).

A researcher with no relationship with the facility and participants conducted semi-structured interviews with the FCs, members of the NH staff, and the NH manager. The interviews were audio-recorded and transcribed. The anonymized transcripts were then reviewed to verify their accuracy. The pre-FCC interviews (FCs *n* = 2, NH staff *n* = 4) aimed to identify the perceived barriers to and facilitators of FCC implementation, while the post-FCC interviews investigated the perceived impact and usefulness of the FCC (FCs *n* = 11, NH staff *n* = 4, research staff *n* = 1).

A member of the research team and the nurse responsible for the FCC collected in-the-field notes over the entire study. The notes helped to integrate the data and provided an audit trail detailing the main turning points throughout the research process. Table 2 shows a summary of the data collection methods with their time points.

#### 2.1.5. Data Analysis

The analysis of the qualitative data (i.e., interviews transcripts and in-the-field notes) consisted of two separate processes. First, an inductive content analysis according to Graneheim and Lundman was applied [36]: (a) familiarization—all qualitative data were read carefully and repeatedly; (b) compilation—two researchers independently examined the transcripts using an open-coding approach whereby the most significant words and phrases (*units of meaning*) were highlighted; (c) condensing—two researchers independently reduced each meaningful unit to a descriptive label (*code*); (d) categorization—the codes were compared and grouped into sub-categories according to their similarities. Second, a deductive approach was adopted by fitting the identified sub-categories in the existing “End-of-life communication theory in nursing homes” [21]. All team members participated in frequent meetings to discuss the codes, how the sub-categories fit in the selected theory, and the illustrative quotations. When the sub-categories did not fit the given theory, they were grouped into new categories, which complemented the original theory. The categories are illustrated by participant quotations, which are identified by an alphanumeric code that indicates the data source. An example of the coding process is detailed in Table 3. ATLAS.ti 8 aided the analysis.

Each element of the existing communication theory was defined as ‘present’ if it emerged from at least one of the data collection sources, otherwise as ‘absent’, regardless of the positive or negative impact on end-of-life communication.

The quantitative data (i.e., clinical records and questionnaires) were summarized as mean and standard deviations. SPSS version 28.0 was used for the descriptive statistics.

#### 2.1.6. Ethics

The Ethics Committee of the University of Torino (Italy) approved the study (Reference 131362, 5 March 2020). All participants gave their written informed consent to participate in the study.

### 2.2. Development of a Situation-Specific Theory of End-of-Life Communication in Nursing Homes

The development of the situation-specific theory of end-of-life communication in NHs followed an integrative approach that included four steps: (1) checking assumptions for theorization; (2) exploring the phenomenon through multiple sources; (3) theorizing (i.e., initiation, process, and integration); and (4) reporting, sharing, and validating the theorization [26].

#### 2.2.1. Checking Assumptions

The first assumption was that diversities and complexities could exist within the phenomenon of end-of-life communication in NHs, and only some of them are represented in the proposed theory. Second, the theory-development process was assumed to be cyclical and evolutionary, and occurred in the specific Italian socio-political, historical, and cultural context, which may limit the application of the theory. Finally, the entire process was supported from a nursing perspective.

#### 2.2.2. Multiple Sources of Theorizing

Multiple combined sources were employed using an inductive approach: (a) reviews of the literature; (b) research findings; (c) clinical experience; (d) experience from a transnational research project; (e) clinical discussions with nurses, physicians, psychologists, NH managers; and (f) discussions with international colleagues (Figure 2).

#### 2.2.3. Theorizing: Initiation, Process, and Integration

The theorizing process of this situation-specific theory is clinically grounded and started with a simultaneous induction from three literature reviews [11,37,38]. Then, the initial theorizing process was validated through interviews with the FCs of NH residents [9,39] and nurses [40] who work in NHs and led to a first theory of end-of-life communication in NHs [21]. The theorizing process moved forward by integrating the experience from a transnational quality improvement project [27] and updated research findings emerged from the implementation of such project in Italy [41]. Finally, knowledge from clinical discussions and discussions with international colleagues was incorporated and a final situation-specific theory was developed (Figure 1 and Figure 2).

Several strategies were employed throughout the entire theorizing process to integrate the upcoming knowledge, including both internal dialogues (i.e., conceptual frameworks, memo writing, journal writing) and external dialogues (i.e., discussions with colleagues, research members, research participants, and participation in seminars, conferences, and panel discussions) [24]. Reflexivity was also instrumental in the process of integration and considered both the Italian socio-political, historical, and cultural context, and personal values and meaning about the phenomenon under study (i.e., end-of-life communication in NHs). A theoretical and analytic decision trail was kept during the development process of the theory [24].

#### 2.2.4. Reporting, Sharing, and Validating Theorization

This article reports a refinement of the existing theory of end-of-life communication in NHs [21].

## 3. Results

All the elements of the existing theory of end-of-life communication in NHs were confirmed [21]. Moreover, in addition to the factors related to the NH environment already described in the existing theory, the role of the family environment and the contextual factors external to the NH emerged (Table 4, Appendix A). Specifically, the family environment affected the FCs’ burden in the decision-making process, while political, historical, social, and cultural factors influenced the quality and timing of end-of-life communication between the NH staff and the FCs. Thus, the existing theory was refined and the foundations for a situation-specific theory were provided (Figure 1).

### 3.1. Healthcare Professionals–Family Caregivers and Healthcare Professionals–Resident Communication

Before the FCC, the FCs were usually not satisfied with the communication received and the HCPs described family meetings as unstructured informative encounters which took place on the FCs’ request or to inform the FCs about care decisions made by the staff. After the FCC, the FCs described communication as clear, timely, and supportive and the HCPs took time for regular, informal encounters to provide updates. The FCs hoped for structured FCCs to become routine.

“They [NH staff] tell me what they want to tell, I don’t know if they tell me everything. I feel that sometimes communication is missed.”/“This clear communication reassured me, now I feel calmer and have clearer ideas.” (Pre- and post-FCC interview, FC1)

“Family meetings take place when we need to inform the family about care decisions we made or on their request.”/“In the weeks and months following the FCC, FCs needed more informal communication and updates. I took the time to continue such encounters.” (Pre- and post-FCC interview, staff member 1)

“Scheduled meetings with the medical/nursing staff would be useful for family who need feedback and should become routine.” (Post-FCC interview, FC4)

### 3.2. Family Caregivers’ Understanding

The NH staff recognized the importance of tailoring communications to the FCs’ educational and supportive needs, as well as to their degree of awareness. The FCs’ understanding about their relative’s clinical conditions and possible care options at the end of life improved after the FCC.

“My mum’s illness is progressing, in my opinion, no important decisions need to be made.”/“I did not know that hydration may be enough when she stops eating and we can avoid the need to insert a feeding tube.” (Pre- and post-FCC interview, FC1)

“The FCC made me reflect on things that one unconsciously knows, provided me with awareness of what could happen, and helped me to understand the pros and cons of the choices. One often tends to bury one’s head in the sand while saying ‘there is time’. Reading the booklet and then discussing it with M. made me open my eyes earlier.” (Post-FCC interview/FC6)

“Family meetings allow us to answer family doubts, provide further explanations if necessary, and promote awareness about the pathophysiology of the disease and possible complications.”/“The relative realized that aggressive treatments do not make sense.” (Pre- and post-FCC interview, staff member 3)

### 3.3. Shared Decision-Making between Healthcare Professionals and Residents/Family Caregivers

After the FCC, the FCs usually felt more involved in the decision-making process, became more proactive in confronting the staff, and their trust increased. In some cases, the FCs declined decisional authority while feeling reassured about transferring the decision-making responsibility to the staff because they had the opportunity to share their care preferences and the goals of care were agreed. Instead, when trust was lacking, the FCs feared that the HCPs did not make the best care choices for their relative.

“My mum cannot decide anything, others always decide for her. Sometimes I think, ‘Will they make the right or wrong decisions?’.” (Pre-FCC interview, FC1)

“I’m calmer now because they know what I think and I know what they think. We have agreed on the path to follow.” (Post-FCC interview, FC4)

“Following the FCC, FCs feel much more involved in decisions and more emotionally supported. In their interviews, FCs often state, ‘When I’ll have to make a decision, I know I can trust them, I know they’ll give me the right advice and I’ll follow what they tell me.’” (Post-FCC interview, research staff)

Also, the NH staff found the FCCs beneficial in strengthening relationships with FCs and promoting shared decision-making:

“It has been a wonderful piece of work […]. FCs felt recognized as caregivers and familiar relationships based on mutual respect and with stronger bonds than before were established.” (Post-FCC interview, staff member 3)

“FCCs allowed FCs to be engaged in end-of-life decisions, provided space for sharing and communication, thus enriching the end-of-life experience of FCs.” (Post-FCC interview, staff member 2)

### 3.4. Residents’ Preferences Known

The residents’ preferences for end-of-life care were poorly known by both their FCs and the NH staff, and the latter recognized that there is huge room for improvement to elicit such preferences and the benefit of FCCs. The FCC was an opportunity for the FCs to reflect on their own and their relative’s care preferences and to discuss such preferences with the staff. 

“When people with dementia transition into NHs, they are often no longer cognitively competent, we cannot explore their care preferences anymore, and they usually had not been asked earlier ‘What would you want if that happened to you?’. I see this as a very critical issue and we need to work on this to provide goal-concordant care when the person cannot express their preference anymore.”/“The project allowed us to give voice to the preferences of the residents by promoting reflection among their FCs. This guided the adjustment of the care plan and the provision of care that is potentially consistent with the residents’ preferences.” (Pre-FCC and post-FCC interview, staff member 1)

“My mum has always been a strong woman, and I think that this is no longer a way of life for her. She is vegetating on a bed [...] I wonder if my mum was still cognitively competent, stuck in a bed, what would she want to do? As we have known her, my sister and me, I don’t think she would like to go on.” (Post-FCC interview, FC3)

“I never asked my mum about this and she never brought it up. She has always been a combative spirit, full of energy, but she has dramatically deteriorated in a short time […]. I cannot perceive to what extent she will want to be attached to life.” (Post-FCC interview, FC6)

### 3.5. Family Caregivers’ Preferences Known

The FCCs promoted the reflection of FCs on their own preferences for their relative’s care. The FCs became aware of and shared their care preferences with the HCPs who could support the FCs’ choices. The FCs generally desired palliative-oriented care.

“The meeting clarified things for me and now both the facility and I know which decision I’ll make [...]. I want to limit my mum’s suffering as much as possible and ensure she dies peacefully. I made this clear with the staff and they agreed with me.” (Post-FCC interview, FC4)

“The project does not allow us to elicit residents’ preferences, but gives their FCs the opportunity to reflect both on their own care preferences and on their relative’s potential ones. This reflection then makes the FCs interact with the staff with a different awareness.” (Post-FCC interview, research staff)

The HCPs reported that the FCs’ care preferences heavily guided the goals of care; some FCs wanted to control symptoms and improve their relative’s remaining quality of life, while others wanted to leave nothing undone. However, the FCCs were often the first and late opportunity to explore the preferences of the FCs.

“Some FCs tell you ‘I want to carry my mum to the hospital if she gets worse, I want to do everything possible [...]’. Instead, others called for a painless dignified death.” (Pre-FCC, staff member 3)

“FCCs provided space for shared reflection on topics that FCs often have never been faced with before and are reluctant to engage in.” (Post-FCC, staff member 2)

### 3.6. Contextual Factors

Contextual factors emerged at three levels: the family, the NH, and external to the NH.

#### 3.6.1. Family Environment

When the FCs supported each other and the treatment choices were shared, the FCs did not feel alone, the burden of decisions was shared, and the decision-making process was easier.

“My sister and I are in sync, perhaps because we know our mum well. We move forward and make decisions together as we have always done.” (Post-FCC interview, FC3)

“My daughter and I both read the information the NH provided us. She helps me a lot with these things, she is very sensitive. Then, we discussed it and participated together in the FCC. My brothers also read the booklet: at first, they told me that it was not the case, that there was time; then, when I told them what we had discussed in the FCC they said it sounded good.” (Post-FCC interview, FC6)

#### 3.6.2. Nursing Home Environment

Qualitative data described this NH as a young organization, open to change, eager to improve resident care, with end-of-life communication among its priorities, and with welcoming FCCs. The NH environment was described as familiar and based on availability and trust.

“This is a young facility, we are still running-in. This can be an advantage since practices have not yet been fully consolidated, there is some possibility to sow change.” (Pre-FCC interview, staff member 2)

“We try to establish collaborative and trusting relationships with family caregivers. We want a familiar atmosphere.” (Pre-FCC interview, staff member 3)

“I don’t worry, if something happens I know I can ask and they [NH staff] will answer. I trust them.” (Post-FCC interview, FC4)

“We want to improve the quality of care we provide. Thereby, we are interested in joining projects and getting trained on whatever topics can help us to reach this mission.” (Post-FCC interview, NH manager)

#### 3.6.3. Political/Normative, Historical, Social, and Cultural Factors External to the Nursing Home Environment

##### Political/Normative Factors

Regional regulation that establishes the amount of caring time provided by HCPs according to the intensity of care of residents was mentioned. Particularly, the interviewees judged the amount of caring time that should be guaranteed according to regional law as insufficient and perceived the limited human resources as a threat to high-quality communication. The shortage of staff made some FCs perceive poor involvement in the decision-making process and the provision of standardized care.

“The NH staff are always in a rush.” (Post-FCC interview, FC1)

“I realize how my mum is doing depending on the number of tubes, oxygen, drip, or catheter. I ask for information only when I meet a nurse by chance [...]. My perception is that they are very understaffed.” (Post-FCC interview, FC3)

“There is consensus that FCCs improve the quality of care and family satisfaction with the care and the support received. FCCs need to become routine, we cannot deny persons something with proven efficacy. However, if we really want to improve NH care, the responsibility should not be left to the individual facility, the individual NH manager, or the individual nurse’s good will, but there should be broader supportive guidelines at the regional or national level that adjust current NH staffing. NHs should be given more staff: this would be a tangible signal that communication is recognized as time of care.” (Post-FCC interview, research staff)

##### Historical Factors 

The COVID-19 pandemic has negatively affected the historical staff shortage and high nursing turnover that are typical of NHs and the opportunity to establish meaningful staff–family relationships. Indeed, the FCs perceived poor involvement in their relatives’ life and care decisions as a result of visitation restrictions.

“The difficulties in recruiting personnel and the almost total turnover of the nursing personnel are clear indicators of the ongoing transformation that has been taking place over the last year [...]. The pandemic did not allow us to establish trusting relationships between staff and family.” (In-the-field notes)

“When we come to visit, we are now locked in the room, you do not see or talk to anyone [...]. At this moment, we are not much involved in care decisions.” (Post-FCC interview, FC3)

“The project has been useful to partially recover trusting relationships which had been lost during the pandemic. Visitation restrictions made FCs feel poorly informed and involved in care decisions. These family meetings represented the starting point to recovering relationships.” (Post-FCC interview, staff member 1)

##### Social Factors

Values such as a sense of responsibility and filial duty influenced the desire of FCs to be involved in care decisions after their relative transitioned into the NH.

“My sister and I have a profound sense of duty towards our mum. We have not abandoned her after she transitioned into this home, we want to be present. We act this way probably because we grew up with these values.” (Post-FCC interview, FC3)

##### Cultural Factors

Cultural values such as the taboo of the end of life and death hindered end-of-life communication and knowledge of the residents’ and the FCs’ care preferences. These cultural values were also deeply embedded in the family culture. Some FCs were not even able to pronounce the word “death” and used indirect periphrasis. Moreover, the residents usually did not discuss their end-of-life care preferences with their FCs when they were still cognitively competent.

“I want to avoid hospitalization and desire my mum to be accompanied, you understand for what.” (Pre-FCC interview, FC2)

“Within our family, we are not used to discussing such topics. Our mum has never explicitly spoken to us about these issues.” (Post-FCC interview, FC3)

“FCs are usually reluctant to engage in care conversations because they do not want to think about serious decisions they will be asked to take for their relative.” (Post-FCC interview, NH manager)

“This study allows us to reflect and discuss topics that are usually pushed away and denied.”(In-the-field notes)

“The end of life is often a taboo within the family. Children do not discuss these issues with their parents while they are still cognitively competent. Thus, at the end of life, they are faced with making decisions based on what they think the parent would have wanted. Anyway, it’s always a guessing game, which comes with a significant emotional and decision-making burden.”(Post-FCC interview, research staff)

### 3.7. Residents- and Family Caregivers-Related Care Outcomes

Before the FCC, no residents had any documents completed about care preferences; over the 12 weeks following the FCC, an Advance Decision to Refuse Treatment and power of attorney were signed for eight and two residents, respectively. The median time for a decision after the FCC was 36 days (range 3–88 days) (Appendix B).

Over the 12 weeks prior to the FCC, 7 out of 11 residents used hospital-based services (emergency department access = 3, hospital admission = 2, outpatient department (OPD) access = 5). The reasons for emergency department access were worsening of general clinical conditions or cognitive symptoms. One access led to hospitalization. The other hospital admission was due to dehydration with a 10-day stay. The reasons for outpatient department access were: care for a pressure injury not treatable in the NH (wound care OPD), worsening of cognitive symptoms (Center of Cognitive Disorders and Dementias), pyelostomy replacement (urology OPD), drug-induced gynecological hemorrhage (gynecological OPD), and annual pacemaker control visit (cardiology OPD). In the 12 weeks following the FCC, a resident accessed the urology OPD once for a pyelostomy replacement.

On average, the decisional conflict perceived by the FCs pre-FCC and post-FCC was 1.9 (1.2) and 1.0 (0.8), respectively; perceived care did not change (Appendix B).

## 4. Discussion

This study developed a situation-specific theory of the role of end-of-life communication in contributing to the transition to palliative care in NHs by refining an existing theory [21]. The current literature describes several benefits of end-of-life discussions in NHs at the level of the resident, the FC, and the healthcare system. At the resident level, end-of-life communication was associated with reduced non-beneficial medical care near death and enhanced goal-concordant care. At the FC level, end-of-life communication reduced decisional uncertainty, burden, and emotional distress, and increased FCs’ satisfaction with the care their relative received. At the health system level, end-of-life communication reduced healthcare and hospital costs [11,42]. Recently, timely and frequent end-of-life communication between HCPs and FCs has been proposed as a solution for high-quality palliative care in life-limiting conditions such as dementia [43]. However, knowledge on how end-of-life communication can promote the transition from curative-oriented care to palliative-oriented care in severely ill people is still limited [9,21,38]. This article attempts to explain the modalities by which end-of-life communication contributes to changing care goals in the context of NHs while considering the complex political, historical, and socio-cultural system where end-of-life communication takes place and that may influence the pattern of communication.

Our findings confirmed that the NH environment may influence the timing and quality of end-of-life communication, that in turn impacts end-of-life care by affecting (a) the FCs’ understanding; (b) the shared decision-making between HCPs and residents/FCs; and (c) the knowledge of residents’ preferences and (d) FCs’ preferences for end-of-life care [21]. Moreover, the impact of political/normative, historical, social, and cultural factors on the characteristics of communication has been highlighted.

The FCs perceived ameliorations in the quality of communication with staff after the FCC with regard to both the information and the support received. The FCs felt reassured and described the communication as an “outburst, discussion of relief” and when “timely, regular, and thorough”, the FCs had “clear ideas about the choices to be made”. Regular communication allowed the acquisition of information step-by-step and the gradual development of awareness of a relative’s clinical conditions, which is essential for shared decision-making [37].

Our findings suggest that family meetings improved the FCs’ understanding about their relative’s prognosis and clinical course and made them aware of the opportunity to avoid invasive procedures such as feeding tube insertion. There is a large body of evidence highlighting that when FCs were aware of their relative’s poor prognosis, they preferred palliative-oriented care [16,44,45], instead, lack of awareness was associated with a low perceived quality of life for relatives when active treatment was not provided [46]. We found that family meetings promoted shared decisions by increasing mutual trust and fostering the development of FCs’ proactive attitudes in interacting with HCPs. Therefore, structured communication appears to work on a double pathway, by promoting FCs’ empowerment in addition to strengthening FCs–staff relationships. Our findings confirm the potential of end-of-life communication to promote partnership between FCs and HCPs [37]. Unfortunately, the literature suggests that end-of-life communication usually has an informative intent rather than being aimed at promoting true shared decision-making: 90% of the decisions to withdraw or withhold treatments in residents with advanced dementia were communicated to the FCs post-facto, but only half were discussed before being implemented [47]. This case study adds further insight to the shared decision-making process by highlighting the internal dynamics of the family unit and the role of a supportive family environment in reducing the FCs’ decisional burden when the care preferences of the relative are not known.

Family meetings emerged as an opportunity for the FCs to reflect on their care preferences for their beloved and on what their relative would do if they could still decide. This reflective process promoted awareness of their relative’s preferences and sharing with HCPs. Although knowledge of users’ care preferences is essential for goal-concordant care, which is a quality indicator of palliative care, the preferences of our residents were poorly known, and their FCs’ preferences were often explored too late. These findings confirm the data of a large cross-sectional analysis involving 322 NHs in six European countries, which found that only one-third of residents in Belgium had a ‘do not transfer to hospital’ advance directive, while barely anyone had similar documentation in Italy [14]. Similarly, our findings are close to those of previous focus group interviews with the FCs of Norwegian NH residents who often did not know their relative’s wishes when decisions had to be made and experienced decision-making burden [48]. Poor knowledge of the residents’ preferences is recognized to increase the likelihood of intensive care [14]. Instead, when the preferences were known, they were generally comfort-oriented and directed the HCPs towards palliative-oriented care [49].

As aforementioned, beyond framing end-of-life communication within the NH environment, the main novelty of this paper is to contextualize end-of-life communication in a complex political/normative, historical, and socio-cultural system. This case study took place in a context characterized by a familiar atmosphere, based on availability, trust, and openness to change, which set communication among its priorities. Previous authors found that the institutional culture may influence HCPs’ attitudes surrounding end-of-life decisions [50] with different outcomes for residents [51]. Our interviewees depict the care environment as “familiar”, based on “collaborative relationships with FCs” and with the vison to invest in training to improve the care provided. This suggests a conducive atmosphere and sensitivity to palliative and end-of-life research and highlights the crucial role of leadership in sustaining quality improvement projects and improving the quality of care [52]. This NH is embedded in an Italian long-term care sector suffering from chronic staff shortages that worsened during the pandemic, since it has become even harder to recruit and retain nurses due to the extra nurses being called into hospitals with higher salaries to deal with the care pressure of the pandemic [53]. A previous study conducted in six Piedmontese NHs during the first and second wave of the pandemic found on average 5.7 FTE nurses for every 100 NH beds who had to care for severely ill residents [54], which is even lower than the Piedmontese staffing standard [32]. Our FCs perceived the NH to be extremely short-staffed with nurses always in a rush; the staff shortages and turnover resulted in fragmented communication without a contact person that the FCs could rely on for updated information on their relative’s condition and poor FC involvement in treatment decisions. Moreover, visitation restrictions to limit the spread of COVID-19 compulsorily shifted in-presence communication to remote communication, thus making the FCs often feel outside of their relative’s life and care [31]. Our findings also highlighted the impact of social obligations such as a sense of filial duty on attitudes and involvement in decision-making at the end of life. Italian adult children felt a responsibility to care and advocate for their relative’s care preferences, since parents were the primary caretakers with whom children developed their first life-experiences of trust, security, and affection [55]. Finally, taboos about death and dying hindered end-of-life communication and were responsible for poor knowledge of care preferences. In traditionally Catholic countries such as Italy, HCPs often delay the discussion of such topics for fear of hurting patients’ feelings and destroying hope [56]. Moreover, people often transition into NHs with limited cognitive capacity and only a few have discussed their care preferences with their FCs in advance [49].

Despite the limited sample size, this case study provided an in-depth, comprehensive view of communication in NHs at the end of life based on several data sources and perspectives.

## 5. Conclusions

This case study has highlighted the influence of political/normative, historical, social, and cultural factors in addition to the NH environment on the quality and timing of end-of-life communication, thus improving the existing theory and developing a more thorough, complex, situation-specific theory. Future studies may verify this theory by employing quantitative methodologies on a larger scale.

This situation-specific theory has several implications for research, practice, education, and policy. It can be used to explore new modalities of communication, such as remote modalities, to ameliorate the quality of end-of-life communication and the quality of care. Indeed, aside from the COVID-19 pandemic, a growing number of FCs act as long-distance caregivers and may benefit from remote communication to ease interactions with HCPs and improve access to information. Moreover, this theory can offer guidance to HCPs in addressing FCs’ potential social obligations and cultural taboos during end-of-life conversations that may hinder the discussion of treatment options, thus reaching an in-depth, genuine understanding of preferences and arranging care accordingly. Additionally, it provides NH managers with environment design-related indications to favor a home-like atmosphere and supportive end-of-life communication. Furthermore, this theory can help with structuring communication skills interventions to train HCPs in raising and sustaining these sensitive conversations to promote FCs’ understanding and elicit end-of-life care preferences. Finally, it may be a valuable resource for policymakers when decisions about the allocation of resources in the healthcare sector need to be made, as well as when quality improvement projects need to be implemented and sustained over time in the NH setting.

## Figures and Tables

**Figure 1 ijerph-20-00869-f001:**
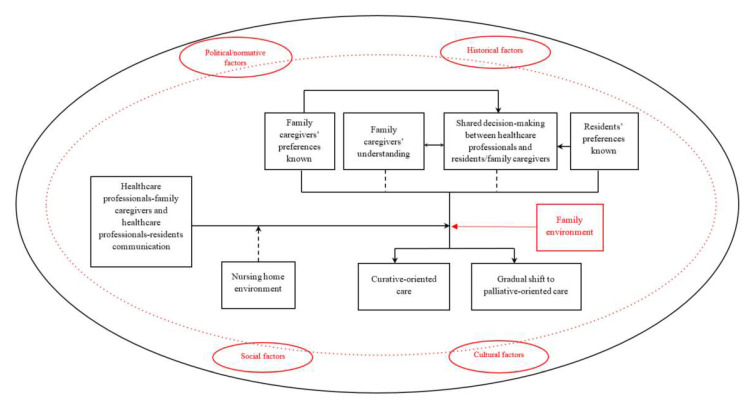
Existing End-of-Life Communication Theory and Refined Situation-Specific Theory of End-of-Life Communication in Nursing Homes. *Note*. Changes to the existing end-of-life communication theory [21] are shown in red. The nursing home environment may influence the timing and quality of end-of-life communication that in turn impacts end-of-life care by affecting (a) the family caregivers’ understanding; (b) the shared decision-making between healthcare professionals and resident/family caregivers; and (c) the knowledge of residents’ and (d) family caregivers’ preferences for end-of-life care. Family understanding and shared decision-making appear not to be essential for the provision of palliative-oriented care, although they have a positive role. Political/normative, historical, social, and cultural factors influence the communication process.

**Figure 2 ijerph-20-00869-f002:**
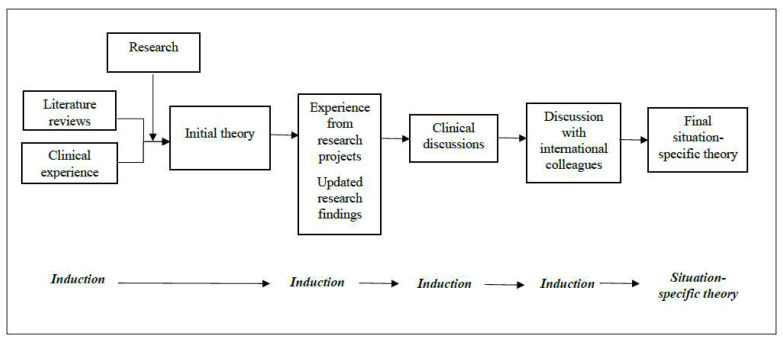
Theorizing Process of the Situation-Specific Theory of End-of-Life Communication in Nursing Home.

**Table 1 ijerph-20-00869-t001:** Main Interview Questions for Family Caregivers, Nursing Home Staff, the Nursing Home Manager, and the Research Staff According to the Time Point.

Time Point	Family Caregivers	Nursing Home Staff	Nursing Home Manager	Research Staff
**Pre-family care conference**	Having described the intervention to you, are there any barriers to its implementation that you would anticipate? Are there any things that you feel will help the process? How do family caregivers currently participate in the development of treatment plans? Can you explain what it has been like for you making decisions about your relative’s care plans?	Can you identify any barriers or facilitators for implementing the intervention? Do you think the intervention will work well for residents and family caregivers? Why? Do you think staff will feel able to accommodate the intervention into their current workload? Why?	Do you think this intervention aligns with the priorities of your care home? Why do you feel this? Do you think the staff in the care home will feel able to accommodate the delivery of the intervention into their current workload? Please describe any barriers that you anticipate would hinder implementing the intervention in your care home	
**Post-family care conference**	How well did the family care conference help you understand the choices and decisions that needed to be made? How well do you feel the nursing home staff recognize your role as a decision maker?	How do you feel family caregivers have responded to the intervention? What do you find meaningful about the intervention for yourself? Compared with your expectations of how the intervention would be implemented, can you explain any challenges that you faced? Do you think the intervention could fit into the normal running of a care home?	Can you describe any barriers or facilitators that you faced in implementing the intervention? Were there any surprises about the intervention or the process? Do you feel that the intervention met the resident’s/family caregivers’ needs?	Did you find the intervention difficult to implement, compared with your expectations of how the intervention would be implemented? Do you feel the caregivers welcomed the intervention? What recommendation would you make with regards to future work in this area?

**Table 2 ijerph-20-00869-t002:** Data Collection Summary with Time Points.

Data Collection Methods	Time Point	
	Before Family Care Conference	After Family Care Conference
Residents’ clinical records (N)	11	11
Family caregivers’ self-administered questionnaire (N)	11	11
Family caregivers’ semi-structured interviews (N)	2	11
Nursing home staff semi-structured interviews (N)	3	3
Nursing home manager semi-structured interview (N)	1	1
Research staff semi-structured interview (N)	-	1
In-the-field notes	Collected	Collected

**Table 3 ijerph-20-00869-t003:** Analytical Process Performed: an Example.

Inductive Approach			Deductive Approach
Units of Meaning	Codes	Sub-Categories	Categories
A FC said: “My mum cannot decide anything, others always decide for her. Sometimes I think ‘Will they make the right or wrong decisions?’.”	Doubting that HCPs make the best care choices	Level of trust	Shared decision-making between healthcare professionals and residents/family caregivers
A FC said: “I’m calmer now because they know what I think and I know what they think. We have agreed on the path to follow.”	Trusting the HCPs		
A research staff member said: “Following the FCC, FCs feel much more involved in decisions.”	Feeling involved in decisions	Family involvement in end-of-life care decisions	
A HCP said: “FCCs allowed FCs to be engaged in end-of-life decisions, provided space for sharing and communication, thus enriching the end-of-life experience of FCs.”	Providing FCs space for discussion		
A research staff member said: “Following the FCC, FCs feel […] more emotionally-supported.”	Feeling emotionally supported	Establishing a partnership between HCPs and FCs	
A HCP said: “It has been a wonderful piece of work […]. FCs felt recognized as caregivers and familiar relationships based on mutual respect and with stronger bonds than before were established.”	Strengthening relationships with FCs		

*Abbreviations.* FC, family carer; FCC, family care conference; HCP, healthcare professional.

**Table 4 ijerph-20-00869-t004:** Elements of the Existing End-of-Life Communication Theory and Contextual Factors that Emerged in the Interviews Before and After the Family Care Conferences.

	Before Family Care Conference N = 6 n/N	After Family Care Conference N = 16 n/N
Elements of the existing end-of-life communication theory		
Healthcare professionals–family caregivers and healthcare professionals–residents communication	6/6	16/16
Family caregivers’ understanding	4/6	16/16
Shared decision-making between healthcare professionals and residents/family caregivers	6/6	16/16
Residents’ preferences known	4/6	8/16
Family caregivers’ preferences known	2/6	16/16
**Contextual factors**		
Family environment	3/6	13/16
Nursing home environment	3/6	14/16
Contextual factors (i.e., political/normative, historical, social, and cultural factors)	5/6	14/16

## Data Availability

Data are contained within the article.

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
