# Peer review of "A Situation-Specific Theory of End-of-Life Communication in Nursing Homes"

_ijerph, 2023, doi:10.3390/ijerph20010869_

Round 1

Reviewer 1 Report

Thank you for the opportunity to review this manuscript. It touches upon an important communication issue (EOL) and yet does not occur often within families and with HCPs. I agree with author(s) arguments that a situation-specific theoretical basis is needed to better situate EOL and its related issues in its particular cultural, social, and political contexts. Author(s) did a good job articulating in the introduction in terms of providing a rationale for the study. That said, I have serious concerns in terms of how literature was structured, analyses were conducted, and discussion framed. One major critique I have is that this study did not develop the theory; it helped inform future studies focusing on data collection on a larger scale using quantitative methodologies. Theory building requires a much more research endeavor than a single case study.  

I am outlining my comments below and hopefully, they are constructive and helpful for future revision.

I would suggest expanding the introduction to include some in-depth literature review of the studies focusing on EOL in nursing homes, and possibly in Italian contexts. It would help readers understand interviewees’ comments and perceptions in this study better. Research on EOL and in collective cultures such as Italy is robust. Author(s) need to demonstrate a good understanding of the existing research, and how it leads to the rationale of the current study.

Significant revision for the Method section is needed. Specifically,

·      I do not believe information before study design is needed. This is not a textbook on theory building. Instead of explaining “how theorizing happened,” maybe explain the context of the study (i.e., the facility; which is in the discussion now).

·      Maybe I am not reading this section correctly, but I am very confused with 2.2 – existing theory of EOL communication in nursing homes. What exactly is this existing theory? It does not have much explanation under this section, but it was being referenced frequently in the paper. In addition to what it is, what are the shortcomings of this theory that author(s) feel compelled to develop another one?

·      Please clarify what does an “instrumental” case study mean? What is the difference between primary study vs. current study?

·      What is FCC exactly? Was it a structured meeting between FCs and HCPs? More details about this are needed.

·      Please explain the scales used. Saying that the decision conflict scale investigates perceived decision conflict does not mean anything at all. Repeating the names of the scales is not an explanation of what they are.

·      On page 5, line 195, it says that the pre-FCC interviews included 2 FCs, but the post-FCC interviews included 11 FCs. Why?

·      Author(s) need to provide more details about the analyses. For example, what were the interview questions (Were they the ones listed in Appendix 2?)? if yes, how did you decide on these questions? Were the staff and the FCs asked the same questions? On average, how long was the interview with each participant? How many pages of transcript were produced? How were codes developed? Any theoretical basis that guided the coding processes?

·      The sample size is too small for meaningful statistical analyses. Only descriptive statistics are possible. Please refrain from claiming any other types of analyses in the paper.

Results

·      I am not sure if the explanation and the exemplars under historical factors was sufficient. The exemplars are mainly about the impact of the pandemic.

·      Is the culture factor about family culture as well as the Italian culture?   

Discussion

·      In addition to moving the paragraph on the Italian long-term facilities (page 12, line 488) to the Method section, the discussion can focus on current findings against the findings from previous studies, in Italian contexts or in other contexts. In other words, are similar findings found from this study? If anything different, why? How did the current study support the existing body of research?  

·      Please talk about the contributions of the current study in the discussion. All the findings seem to be consistent with many other studies addressing challenges of EOL and benefits of having such discussions with HCPs when given opportunities. Therefore, it is difficult to assess the unique contributions of the current study in the larger body of research on this topic.

·      I will also suggest author(s) spend more time talking about possible future studies that can be done based on the findings of the current study. Or give a better discussion of the “theory” they claimed to have developed.

Reviewer 2 Report

In my opinion this is a very interesting and sophisticated research paper and I would recommend to publish this article.

I only have some minor questions or suggestions for changes: 

1.) page 5: Why have only 2 FCs been interviewed before the FCC ( compared to 11 FCs after the FCC) ? Is there an explanation for this number ?

2.) Data analysis (page 5/6): In my opinion, some more explanations concerning the coding procedure would be nice. Would it be possible to offer some examples for the extracted "key descriptive content" and its transformation into codes ? 

3.) page 11: The comparison of hospital services used before the FCC and after the FCC is a little bit problematic, I would say. Perhaps it would make more sense to restrict this comparison to particular kinds of hospital services used. By the way, this is an open question, but I feel a little bit uncomfortable with the implication that the reduction of all hospital services after an improvement of communication would be a good thing (e.g. "care of a pressure injury not treatable in NH "). 

Reviewer 3 Report

The results section is hard to read. I did not clearly find how many patients/caregivers/nurses were interviewed or how many charts were consulted

Otherwise this is a nice qualitative study. I suggest to add the study design to the title: qualitative interview study and case study 
